# Chronic Exposure to the Food Additive tBHQ Modulates Expression of Genes Related to SARS-CoV-2 and Influenza Viruses

**DOI:** 10.3390/life12050642

**Published:** 2022-04-26

**Authors:** Krisztina Németh, Peter Petschner, Krisztina Pálóczi, Nóra Fekete, Éva Pállinger, Edit I. Buzás, Viola Tamási

**Affiliations:** 1Department of Genetics, Cell- and Immunobiology, Semmelweis University, 1089 Budapest, Hungary; nemethkriszti9502@gmail.com (K.N.); kikiczy@gmail.com (K.P.); fekete.nora@med.semmelweis-univ.hu (N.F.); pallinger.eva@med.semmelweis-univ.hu (É.P.); edit.buzas@gmail.com (E.I.B.); 2Department of Pharmacodynamics, Semmelweis University, 1089 Budapest, Hungary; petschner.peter@pharma.semmelweis-univ.hu; 3MTA-SE Neuropsychopharmacology and Neurochemistry Research Group, 1089 Budapest, Hungary; 4Bioinformatics Center, Institute for Chemical Research, Kyoto University, Kyoto 611-0011, Japan; 5HCEMM-SU Extracellular Vesicle Research Group, 1089 Budapest, Hungary; 6ELKH-SE Immune-Proteogenomics Extracellular Vesicle Research Group, 1089 Budapest, Hungary; 7Department of Molecular Biology, Institute of Biochemistry and Molecular Biology, Semmelweis University, 1094 Budapest, Hungary

**Keywords:** tBHQ, SARS-CoV-2, influenza, CD4, T cells, food additive, antioxidant, AHR, NRF2

## Abstract

Background. *tert*-butylhydroquinone (tBHQ) is an antioxidant commonly used as a food additive. Studies suggest that tBHQ could modulate immune responses to influenza and SARS-CoV-2 infection. In our transcriptomic analysis we explored the molecular mechanisms behind tBHQ’s modulatory properties and the relationships to respiratory viral infections. Methods. tBHQ was administered *per os* to BALB/c mice (1.5% [*w*/*w*]) for 20 days. Splenic T cells were isolated with magnetic separation and subjected to transcriptomic analysis. Gene-set enrichment analysis and g:Profiler was conducted to provide a functional interpretation of significantly changed genes. Further analysis for AHR/NRF2 binding sites was performed with GeneHancer. Results. In CD4^+^ cells, we found significantly altered expression of 269 genes by tBHQ. Of them, many had relevance in influenza infection such as genes responsible for virus entry (*Anxa1/2*, *Cd14*), interferon signaling (*Dusp10*, *Tnfsf13*), or prostaglandin synthesis (Ptgs1/2). In SARS-CoV-2 infections, interferon signaling (*Ifitm*1), proteolytic enzymes (*CtsB*), and also cell-surface proteins (*Cd14*, *Cd151*) were among the prominent alterations after tBHQ exposure. Of these genes, many had one or more binding sites for AHR and NRF2, two major xenosensors triggered by tBHQ. Conclusions. Our results strongly suggest that a common food additive, tBHQ, can modulate virus-dependent processes in both influenza and SARS-CoV-2 infections.

## 1. Introduction

*Tert*-butylhydroquinone (tBHQ) is a common food additive used as a fat antioxidant to increase the shelf life of various fat products, such as pasta, cereals, nuts, margarines, etc. It serves the same purpose in cosmetics. Its use is increasing worldwide, and data are accumulating about its biological effects [1]. At the same time, several health problems have been linked to the adverse effects of this substance [2]. Since tBHQ is a common food additive, it can be counted as a serious environmental factor and there is constant exposure to it.

Many published articles report that tBHQ modulates the immune response [3,4,5,6,7,8]. Freeborn and coworkers found that mice on a tBHQ diet had a weakened immune response to influenza infection (influenza A/PR/8/34 (H1N1)) and that mice eating a tBHQ-spiked diet were slower to activate both CD4^+^ T and CD8^+^ T cells, resulting in slower clearance of the virus [9]. Moreover, mice were easily re-infected at a later time-point with a different, but related strain of influenza. Mice fed with a diet containing a tBHQ dose equivalent with that of a human diet experienced longer illness and lost more weight. This suggests that tBHQ impairs primary and memory immune responses and impacts the efficacy of vaccines, such as regularly given flu vaccines [9]. 

The constant environmental exposure to tBHQ from food or cosmetics and the correlation of tBHQ intake with the immune response to respiratory virus infections (e.g., influenza A and SARS-CoV-2) inspired us to study the virus-related effects of this substance at a transcriptomic level. 

## 2. Materials and Methods

### 2.1. Animals

Male BALB/c mice (wild type, age between 12 and 13 weeks, 20–30 g) were housed in a normal light-cycle (12 h light/12 h dark) animal care facility and were kept either on a tBHQ-supplemented (1.5% *w*/*w*; Fluka, Burlington, MA, USA) or control diet for 20 days. For feeding, food pellets were prepared as follows: AIN 76-A semi purified diet (MP Biomedicals, Solon, OH, USA) was mixed with 1.5% tBHQ (*w*/*w*), 8% warm distilled water (*w*/*w*) and 3% gelatin (*w*/*w*). After cooling, the mixture was shaped into food pellets. For control animals, a control diet (vehicle) was prepared the same way, but without tBHQ. The dose of tBHQ was similar to that used for synthetic phenolic antioxidants in analogous animal models [10]. The animals were provided water and food ad libitum, and the weight of the animals was monitored regularly (Appendix A). Mice were sacrificed using CO_2_. All animal experiments followed the European Union’s Council Directive (86/609/EEC) and were approved by the Semmelweis University’s Institutional Animal Care and Use Committee.

### 2.2. Transcriptome Analysis

Spleens of tBHQ-fed and control animals were removed, and CD4^+^ cells were isolated with magnetic separation using an autoMACS separator (CD4 (L3T4) Microbeads, Miltenyi Biotech, Germany). By flow cytometry, we found that 90% of the isolated cells were CD4^+^CD3^+^T cells (Appendix A). From T cells, RNA was isolated with RNeasy Mini Kit (Qiagen, Germantown, MD, USA). RNA quality was assessed with Bioanalyzer (Agilent, Santa Clara, CA, USA), and samples were subjected to cDNA microarray analysis using an Agilent platform (Agilent mouse 4 × 44 K arrays; One-Color Microarray-Based Gene Expression Analysis, Low Input Quick Amp Labelling protocol).

Raw microarray data were processed with Agipreprocess 44 × 4 program package, normalized with Limma package using the quantile normalization method. Comparison of genes in tBHQ-fed and normal mouse chow groups was performed with lmfit and eBayes functions. Significance was determined by Benjamini–Hochberg correction (defined as q-value) [11], and changes were considered statistically significant when the generated q-values were below 0.05. Heatmap visualization of gene expression differences was carried out using Multiexperiment Viewer Tool [12,13]. Genes with similar expression patterns were grouped with hierarchical clustering (Euclidean distance, average linkage) [14]. The data supporting the results of this publication were deposited in NCBI’s Gene Expression Omnibus [15] and are accessible through GEO Series accession number GSE99152 (http://www.ncbi.nlm.nih.gov/geo/query/acc.cgi?acc=GSE99152, accessed date: 26 July 2021).

Gene set enrichment analysis was used to determine gene signatures based on GO (Gene ontology) datasets [16]. Normalized enrichment scores (NES) were considered significant if *p* < 0.05, and false discovery rate (FDR) was <0.25. Microarray data of 47 genes (marked with asterisk in Appendix A) were validated with quantitative PCR (qPCR) using Taqman assays (Taqman Array Fast Plate, Applied Biosystems, Waltham, MA, USA). 

We also used g:Profiler to perform statistical enrichment analysis and to provide interpretation of significantly altered genes in different databases (g:Profiler version e101_eg48_p14_baf17f0). The biological databases used were GO; Gene Ontology, KEGG; Kyoto Encyclopaedia of Genes and Genomes; REAC; Reactome; WP; WikiPathways; TF TRANSFAC and CORUM; the Comprehensive Resource of Mammalian Protein Complexes databases. g:Profiler analysis was applied only for those genes that were related to influenza or SARS-CoV-2 virus infection and also were significantly changed in microarray experiments after tBHQ exposure. All mRNAs studied are expressed in CD4+ T cells as listed in the Biogps database (Biogps.org) [17].

GeneHancer is a database embedded in GeneCards. It is a collection of human enhancers and their inferred target genes created using four methods: promoter capture Hi-C, enhancer-targeted transcription factors, expression quantitative trait loci, and tissue co-expression correlation between genes and enhancer RNA. It banks data about regulatory sites of a gene that are derived from CHIP-Seq results. GeneHancer uses the following sources: ENCODE project (ENCODE Project Consortium, 2012); Ensembl regulatory build [18]; FANTOMs atlas for active enhancers [19]; VIST Enhancer Browser [20]; dbSUPER super-enhancers [19]; EPDnew promoters [21]; UCNEbase [22]. During the analysis, we searched for putative AHR (aryl hydrocarbon receptor) and NRF2 (nuclear factor-erythroid factor 2-related factor 2) binding sites on 27 genes that have relevance in influenza or SARS-CoV-2 infections. In the analysis elite AHR or NRF2 binding sites were taken into consideration (GH-score marked with *). GH type and GH identifier were also added to the table.

### 2.3. Statistical Analysis

Beside specific statistical methods used to analyze microarray data, all other results were expressed as the mean ± standard deviation (SD). Two-tailed Student’s *t*-test was used to calculate the statistical significance of differences between groups. *p* values < 0.05 were considered statistically significant. 

## 3. Results

### 3.1. Profiling mRNA Expression after tBHQ Treatment

Comparison of gene expression profiles showed 269 differentially expressed genes in mice fed with tBHQ-containing food as compared to controls (180 upregulated and 89 downregulated genes, Benjamini–Hochberg adjusted value q < 0.05, Appendix A). From them, 98 genes showed changes > 2-fold or <0.6-fold, respectively (Figure 1). Microarray data were validated with Taqman assays with a good correlation (R^2^ = 0.8908, Appendix A). The genes chosen for validation were randomly selected from Appendix A. 

### 3.2. Network Analysis

In addition to expression analysis of the individual genes, we carried out functional analysis with GSEA and g:Profiler. To obtain a complete picture of the gene expression changes associated with tBHQ, we performed ranked gene list analysis using GSEA (gene set enrichment analysis) from the whole gene repertoire present on the array (whole-genome ranked list without using a cut-off). After GSEA analysis, significantly changed genes (Appendix A) were visualized as separated subnetworks based on GO categories using edge cutoff 0.5 (Figure 2). The whole network can be found as a Appendix A. To reduce spurious findings, we chose a restrictive false discovery rate cut-off of <0.25 for selecting enriched gene sets. Based on Figure 2, GO categories can be classified to many functional groups, separating different clusters of the network. Some important clusters are related to Immune response, Leukocyte migration and chemotaxis (Figure 2A,B), Apoptosis (Figure 2C), Endocytosis (Figure 2D), Regulation of cell signaling (Appendix A), and Secretion and transport (Appendix A).

For the g:Profiler analysis, we selected genes that are significantly changed both in the transcriptomic experiments and in influenza or SARS-CoV-2 infection. The g:Profiler analysis was performed using six databases (GO, KEGG, WP, REAC, TF, and CORUM). We found 40 terms that were changed after tBHQ treatment and in influenza. The most important terms are related to Immune response, Antioxidant processes, Detoxification, and Arachidonic acid dependent mechanisms (Figure 3A). Significance values of the terms are shown on Appendix A. 

There were 22 terms that can be related to SARS-CoV-2 infection and are modulated by tBHQ. One term was specific for SARS-CoV-2 infection (*Network map of SARS-CoV-2 signaling pathway*). Other terms were mostly connected to the defence response and viral infection (e.g., *viral entry*, *viral life cycle*, *lytic vacuole*, *lysosome*, *external site of the plasma membrane*). Some of them were related to *immune response*. (Figure 3B; significance values of terms: Appendix A). 

tBHQ as a xenobiotic regulates intracellular receptors that are involved in detoxification, such as NRF2 or AHR [5,6,7]. Search analysis for these two receptor binding sites was performed with GeneHancer of the GeneCard database. We analyzed genes that are significantly changed on the array after tBHQ treatment and have importance in the studied viral infections. Using the database, we found numerous elite AHR or NRF2 binding sites (Appendix A) during promoter-to-gene or enhancer-to-gene association analysis.

## 4. Discussion

In this manuscript, we provide evidence that chronic, 20 day-long exposure to the food additive tBHQ changes gene activities related to immune response, apoptosis, endocytosis, and secretion. In addition, our data also suggest that tBHQ might alter the immune reaction against respiratory virus infections such as influenza or SARS-CoV-2, since tBHQ modulates the expressions of genes that are involved in infection and replication of these viruses [9]. 

tBHQ is a chemical that activates two major xenosensors, AHR and NRF2 in the cell. Activated receptors modulate transcription by binding to the responsive elements/enhancers of tBHQ target genes. Virus infections interfere with NRF2 signaling mechanisms since they cause oxidative stress and provoke immune response, apoptosis, or necrosis. In infection, such as SARS-CoV-2 or influenza, host cells defend themselves against damages caused by the virus with activation of the NRF2-Keap1-ARE pathway. Moreover, NRF2 decreases the expression of ACE2 (angiotensin-converting enzyme 2) in the membrane protecting the cell from virus overload. Due to the above effects of NRF2, compounds that activate this xenosensor could be beneficial in infection and complex antiviral processes [23,24,25,26]. The AHR receptor is also activated in SARS-CoV-2 or influenza infections, initiates antiviral mechanisms and controls multiple aspects of the immune response [27]. 

Since both NRF2 and AHR can be activated by tBHQ, we assumed that the studied genes might have putative binding sites for these two proteins. Many of these genes bind the two xenosensors, but not all of them, as it was verified with GeneHancer analysis. Therefore, tBHQ triggers other signalling pathways in the cell that are independent from NRF2 and AHR.

### 4.1. tBHQ Exposure and Influenza Virus Infection 

Many groups reported that tBHQ binds to the hemagglutinin proteins (HA) of influenza viruses and stabilizes the non-infectious conformation. tBHQ can bind directly to influenza H3, H7 and H14 HAs [28]. Russell and coworkers reported that the tBHQ binding site is located in the hydrophobic pocket formed between two HA monomers. Since HA is a trimer, each of them has one tBHQ binding site. Occupation of this site by tBHQ stabilizes the neutral pH structure that presumably inhibits the conformational rearrangements required for endosomal membrane fusion [29,30,31].

These studies show that tBHQ directly modulates virus entry. Since this food preservative regulates a wild spectrum of genes, our goal was to find indirect connections between tBHQ (i), its responsive genes (ii), and influenza infection (iii). Our study resulted in numerous possible links between the food antioxidant (which we are, unfortunately, constantly exposed to in everyday life from food and cosmetics) and influenza virus infection (for genes see Table 1 and Figure 4). 

tBHQ upregulated mRNA expression of many receptors that are involved in virus entry, such as *Anxa2* and *FPR-RS2* (*Anxa1*) receptors (they bind virus associated Anxa proteins that promote infection of the host cell) and other “minor” receptors, e.g., *Cd14* and *Mgl1* [32,33,34]. Anxa1 and Anxa2 have two different roles during infection. Anxa2 incorporates into the IAV (influenza A virus) envelope to convert plasminogen to plasmin. This serine protease cleaves HA to facilitate virus entry and replication. Anxa1 interferes with virulence in many ways: it facilitates endosomal trafficking of IAV, enhances IAV-induced apoptosis as well as virus replication, and incorporates into the viral envelope during budding of the virus [35,36,37]. 

Genes involved in INF-α/IFN-γ signaling such as the apoptosis inhibitor APRIL (TNFSF13) and DUSP10, that regulates virus entry and replication, are changed by tBHQ [38]. We also found that chronic tBHQ exposure significantly altered activity of many other genes that are part of virus-dependent INF signaling (Table 1). Among upregulated ones, there is the membrane receptor *TREM1* [39], the TLR4 adaptor protein TIRAP [40], and the *Infβ* regulator *Rel* gene [41]. TREM proteins do not have specific ligands, they are activated by danger signals (DAMP). Activation of TREM will lead to chemokine and cytokine production. In lung injuries, such as influenza infection, activated TLR4 receptors increase proinflammatory downstream effects, such as cytokine secretion. Pro-inflammatory cytokines are associated with a worse clinical outcome. In influenza and in some influenza like infections (such as SARS-CoV-2), genetic polymorphism of *TIRAP*, an accessory protein of TLR4, attenuates the function of TIRAP, leading to the reduced production of pro-inflammatory cytokines [42]. 

*Usp38* negatively regulates type I INF expression and is upregulated in isolated T cells [43]. Type 1 INF target genes, such as *Gzmb* and *Otud,* are also modulated; *Gzmb* is upregulated and *Otud* deubiqitinase is downregulated [44,45]. Both IAV infection and tBHQ exposure modulates prostaglandin synthesis [46,47]. Our results show that after chronic administration of this food antioxidant, mRNA expression of *Ptgs1* and *Ptgs2* enzymes were increased in splenic T cells. There is no evidence in the literature that these enzymes are directly regulated by tBHQ, but it is known that this antioxidant activates NRF2, one of the major regulators of prostaglandin synthesis [48]. Therefore, we hypothesized that the high expression of the mentioned genes is due to NRF2 activation by tBHQ. Gene activities of MMP7/9 migratory proteins are also upregulated in influenza infection [49,50]. MMP-7 is known to reduce tissue damage, thus promoting proper lung function during the attempted resolution. MMP-9 has an opposite effect. If levels of the enzyme are high in the lung (where the influenza virus enters), it creates significant tissue damage [51].

### 4.2. tBHQ Exposure and SARS-CoV-2 Virus Infection

Given that tBHQ is part of our daily food intake and its effect on the immune response is well known, we hypothesized that tBHQ might impact the course or outcome of SARS-CoV-2 infection. Most of the effects of tBHQ are mediated by NRF2 and many NRF2 inducers (dimethyl fumarate and 4-octyl itaconate) inhibited the replication of SARS-CoV-2 and decreased levels of the inflammatory response [52].

Among the genes with significantly altered expression upon exposure to tBHQ, there were several that are potentially related to coronaviruses (for genes see Table 2 and Figure 5).

IFITM proteins are expressed in many types of T cells and are responsible for INFγ-dependent T cell activation. Replication and entry of SARS-CoV-2 is restricted by IFITM proteins. IFITMs appear to block viral infection by reducing host membrane fluidity at sites of viral entry [53,54]. Since we found significantly higher expression of *Ifitm1* and *Iftm2* mRNA in tBHQ fed mice, we assume that this antioxidant is protective against SARS-CoV-2 infection and is associated with higher resistance to viral infection. On the other hand, in severe SARS-CoV-2 infection, the immune response is often inadequate and leads to cytokine storm, a condition with a pathological number of T, NK cells and high levels of various pro-inflammatory cytokines, which in some subjects leads to hyper-inflammatory syndrome characterized by a fulminant and fatal hypercitokinaemia [55]. During this life-threatening hyperinflammation, the infected cells show an impaired capacity to produce proteins that play a role in interferon signaling (e.g., IFITM) [56].

The possible protective role of tBHQ during infection is also suggested by the upregulation of *Cxcr2* (Table 2). Chemokine receptors such as CXCR2 (IL8RB) are important factors that determine the severity of coronavirus infections. Blocking the expression of these proteins correlates with a bad outcome of the disease and the inability to control virus replication [57]. Cathepsins of the host cell play an important role in virus infections; one of their functions is to activate virus envelope glycoproteins (e.g., CTSB). The endosomal cysteine proteases CTSL and CTSB mediate cleavage of the SARS-CoV-1 spike-protein, which is necessary for entry of the coronavirus into host cells [58,59]. Our results show that *Ctsb* mRNA is increased after tBHQ treatment, suggesting its modulatory function on virus entry. 

Cell surface proteins involved in virus binding such as *Cd151* and *Cd14* were increased and *Cd209d* mRNA expression was decreased after exposure to tBHQ. CD151 is enriched at the viral entry sites of COVID-19 viruses. Both soluble and membrane-bound CD14 is involved in COVID-19 infections [60]. Membrane bound CD14 recognizes various viral structures such as F-glycoprotein or RNA and activates inflammatory receptors. It is also responsible for inflammasome activation. The fact that bats have defective inflammasome activation may suggest the importance of CD14 in the host defense against SARS-CoV-2 [61]. A soluble form of CD14 has relevance in disease outcome. Its plasma concentration increases markedly with severity of COVID-19 illness [62]. Perforin 1 was also changed after tBHQ treatment. Levels of this protein decrease with age and body mass index and it is non-functional or missing in perphorinopathies. Diminished perforin level correlates with severity of SARS-CoV-2 related illnesses and there is higher risk for SARS-CoV-2 infection [63,64,65,66,67]. There were many other gene activities that were modulated both by SARS-CoV-2 virus and tBHQ, such as *Ppp1r3d*, *Syn2,* or *Osm. PPP1R3D* protein plays a role in apoptotic processes and *SYN2* has sequence homology with SARS-CoV-2, therefore its level might influence immune reaction against viruses. *Osm* increases INFα level and activates immunostimulatory function. The mRNA of these genes were all upregulated after chronic tBHQ exposure [68].

Antiviral *Padi4* mRNA is upregulated after tBHQ exposure. Studies with lung epithelial and adenocarcinoma alveolar cell models revealed that *Padi4* mRNA levels were elevated in SARS-CoV-2-infected NHBE cells (human bronchial epithelial cells). PAD proteins have also been identified as key regulators of cellular extracellular vesicle (EV) release [69]. There is one published case-study that highlights the effectivity of human EVs against SARS-CoV-2 [70]. Therefore, PAD proteins could hypothetically be beneficial in SARS-CoV-2 infection [69].

We were also interested in whether the two xenosensors, NRF2 and AHR (representing known targets of tBHQ in hepatocytes and other cell types), were involved in the regulation of genes of significance in influenza and SARS-CoV-2 infection after chronic exposure to this food additive. We found many putative enhancer sites using GeneHancer database from GeneCard. Most of the genes of significance in viral infections contain one or more *AHR*/*NRF2* binding sites and some of them have both. CtsB, CD151, and April have three *AHR* binding sites, and Osm has three *NRF2* binding sites. Anxa2, Tirap, Rel, or CtsB have both. There were some genes in both groups that did not have any of the searched enhancers such as Cxcr2, CD14 and unexpectedly, Ptgs2 (Appendix A). Their regulation is likely independent of AHR and NRF2 [52].

ACE2 is one of the most important entry receptors of SARS-CoV-2 viruses. If the virus binds to ACE2, it blocks ACE2 activity and thus reduces enzyme expression in the membrane. A diminished amount of ACE2 alters the physiological functions of the enzyme, such as modulation of the angiotensin II-AT1 axis, and its role in immune, antithrombotic, and anti-fibrotic processes. This may lead to severe symptoms, e.g., cytokine storm. On the other hand, epigenetic studies show that after infection with the virus, a substantial hypomethylation occurs in the ACE2 gene. In peripheral T cells of infected immune cells, this will lead to high virus dissemination. In our studies, we investigated ACE2 on a transcriptomic level. Our results do not support epigenetic studies, neither ACE2 enzyme-reduction after virus internalization. In the present microarray analysis, tBHQ did not change the mRNA expression of the ACE2 gene [71,72].

## 5. Conclusions 

Chronic exposure to various chemicals, such as in food, cosmetic additives, pollutants, cigarette smoke nanoparticles, etc., raise the question of how these molecules affect the human body. In our study we investigated the effect of tBHQ, a food preservative, on infections. With our transcriptomic data, we drive attention to the possible modulation of the response against influenza and coronaviruses by our chronic exposure to food additives and antioxidants such as tBHQ. Recent studies show that tBHQ modulates immune responses against influenza infections and directly binds to the HA protein of the virus. Our transcriptomic data added further evidence that tBHQ modulates the infection process of influenza. It modulates mRNA expression of receptors that play role in virus entry (e.g., *Anxa1/2*, *Cd14*, *Mgl1*). tBHQ significantly changes the gene activities of genes that are involved in immune responses against virus infection, such as interferon signaling, prostaglandin synthesis (*Ptgs1/2*), and apoptosis (*Dusp10*). Chronic exposure to tBHQ interferes with other viruses such as SARS-CoV2. Our studies show that surface receptors, e.g., CD proteins (*CD14*), chemokine receptors (Cxcr2), activity of interferon signaling related genes (*Ifitms*, *Osm*), enzymes that activate virus proteins (*CtsB*), mRNA levels of SARS-CoV-2 homologue protein *Syn2* and extracellular vesicle release (Padi4) are all modulated by this compound. 

NRF2 and AHR receptors are highly involved in regulation by tBHQ and their role in virus infections is very significant. We found more AHR and NRF2 binding sites with GeneHancer in the investigated genes, but further in vitro studies are required to ascertain whether they are functional.

Our work provides additional evidence that chronic exposure to tBHQ influences the cellular response to influenza or coronavirus infections. Future studies that test the links between tBHQ and responses to SARS-CoV-2 may even shed light onto the large variation in symptom severity seen in humans. 

## Figures and Tables

**Figure 1 life-12-00642-f001:**
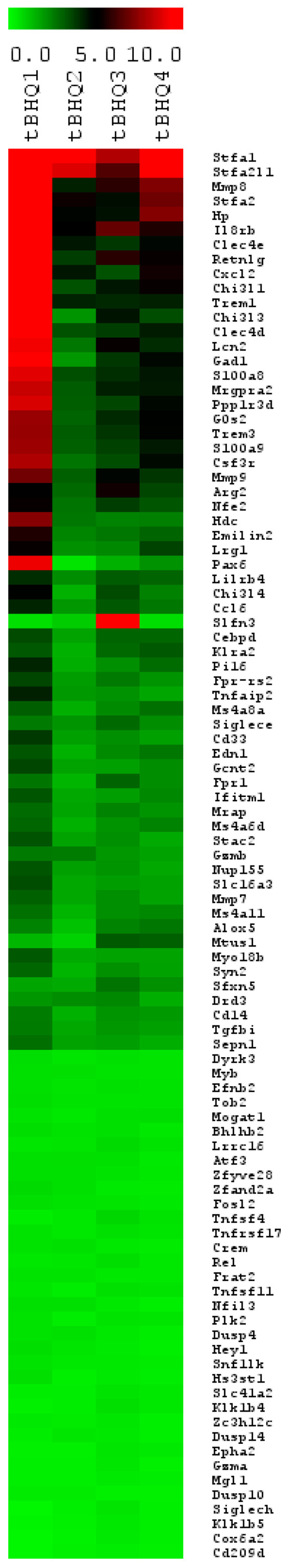
Heatmap analysis of microarray data from isolated splenic CD4^+^ cells after per os tBHQ administration (presented gene expression changes are >2 or <0.6 fold)**.** Heat maps were produced using simultaneous clustering of rows and columns of the data matrix using average linkage algorithm and Euclidean distance metric. The mRNA clustering tree is shown on the left and the sample clustering tree is shown on the top. The colour scale shown at the right illustrates the fold change of the indicated mRNA compared to control: red denotes fold change > 1 and green denotes fold change < 1 (q < 0.05, *n* = 6).

**Figure 2 life-12-00642-f002:**
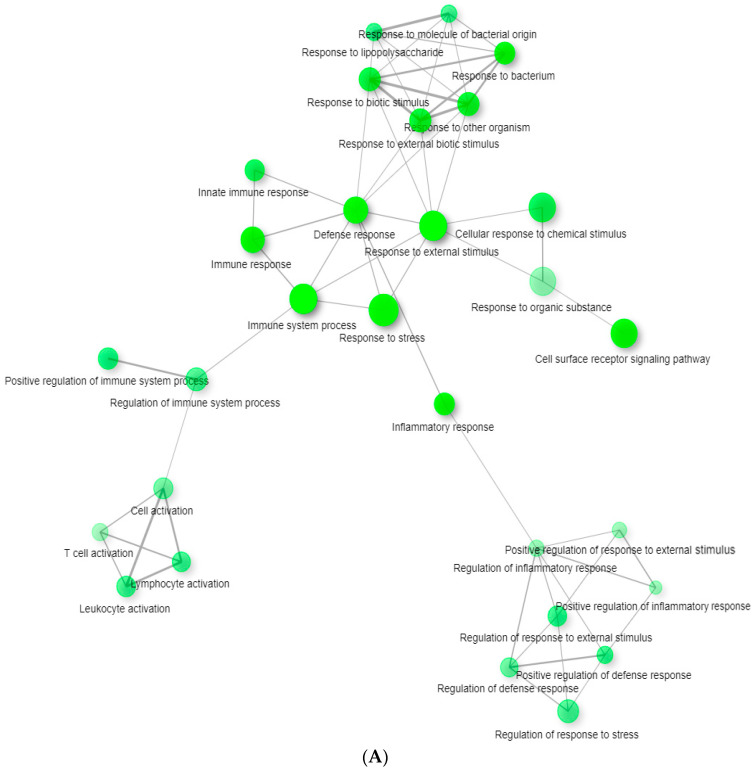
Network analysis of significantly enriched gene sets after per os tBHQ treatment in splenic CD4^+^ lymphocytes. Nodes represent GO pathways significantly changed in gene set enrichment analysis (*p* < 0.05, false discovery rate <0.25). Green nodes represent GO terms, grey edges connection between them. The size of the nodes is proportional with the number of genes in the GO term and the thickness of grey edges represents the number of common genes between two GO terms (FDR = 0.25, *p* < 0.05, edge cutoff 0.5). (**A**) Defense mechanisms and immune response related GO terms. (**B**) Leukocyte migration and chemotaxis. (**C**) Apoptosis related GO terms. (**D**) Endocytosis related GO terms.

**Figure 3 life-12-00642-f003:**
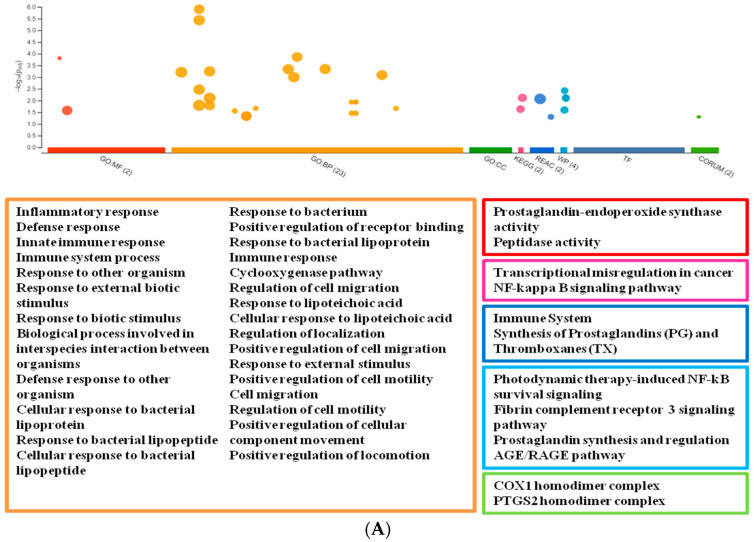
Manhattan plot of enrichment results with g:Profiler. The analysis was done with genes changed in Influenza/SARS-CoV-2 infection after tBHQ exposure. The x-axis represents functional terms that are grouped and colour-coded by data sources (red: GO:MF GO Molecular Functions; orange: GO:BP Basic Processes; green GO:CC Cellular Component; pink: KEGG Kyoto Encyclopedia of Genes and Genomes; dark blue: Reac Reactome; turquoise: WP WikiPathways; indigo blue: TF Transfac; pear green: CORUM Comprehensive Resource of Mammalian Protein). The y-axis shows the adjusted enrichment *p*-values in negative log10 scale. The circle sizes are in accordance with the corresponding term size. The term location on the x-axis is fixed and terms from the same GO sub tree are located closer to each other. Significantly changed terms are represented under the plot in colour frame that is characteristic for the database. (**A**) Influenza (**B**) SARS-CoV-2.

**Figure 4 life-12-00642-f004:**
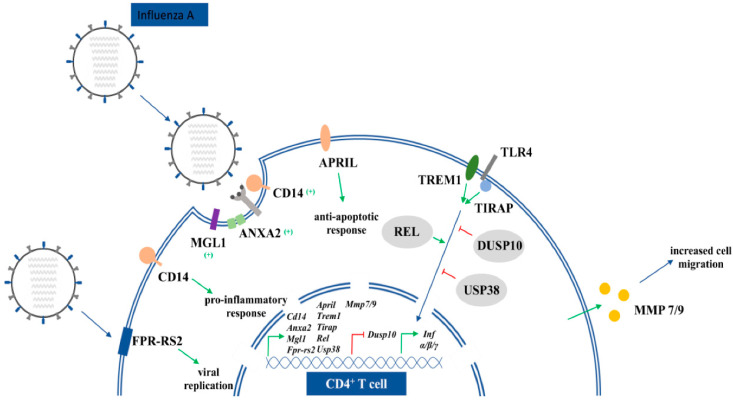
Schematic representation of tBHQ responsive genes that are relevant in influenza infection.

**Figure 5 life-12-00642-f005:**
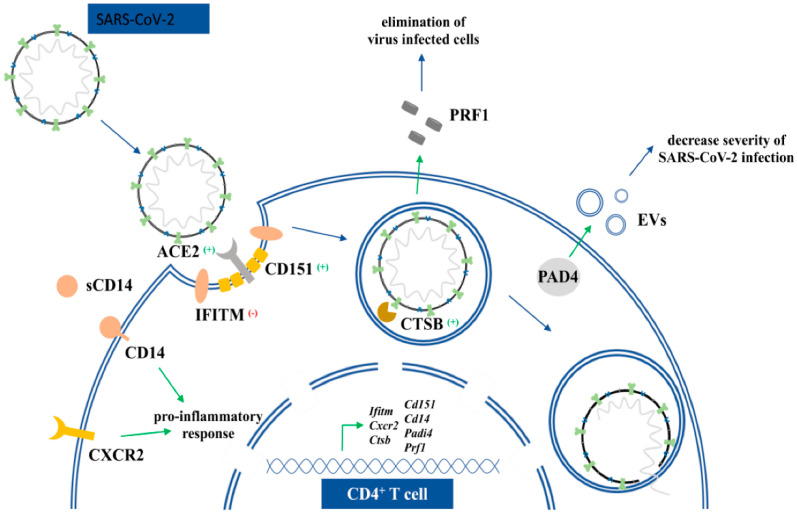
Schematic representation of tBHQ responsive genes that are relevant in SARS-CoV-2 infection.

**Table 1 life-12-00642-t001:** Genes and their relative mRNA expressions that are changed significantly after chronic tBHQ exposure and that are relevant in influenza infection.

Gene Name	Gene Symbol	Fold Change (logFC)	adjP
Annexin A2	*Anxa2*	0.707804	0.019984
Cd14 antigen	*Cd14*	1.154081	0.017954
Dual specificity phosphatase 10	*Dusp10*	−1.41983	0.0107
Formyl peptide receptor, related sequence 2	*Fpr-rs2*	1.407792	0.019984
Granzyme B	*Gzmb*	1.211705	0.042207
Macrophage galactose N-acetyl-galactosamine specific lectin 1	*Mgl1*	−1.41116	0.020827
Matrix metallopeptidase 7	*Mmp7*	1.146066	0.021632
Matrix metallopeptidase 9	*Mmp9*	2.031156	0.0107
OTU domain containing 1	*Otud*	−0.68072	0.021705
Prostaglandin-endoperoxide synthase 1	*Ptgs1*	0.84145	0.037095
Prostaglandin-endoperoxide synthase 2	*Ptgs2*	0.89212	0.026521
Reticuloendotheliosis oncogene	*Rel*	−0.85632	0.048081
Toll-interleukin 1 receptor (TIR) TIR domain-containing adaptor protein (Tirap)	*Tirap*	0.777524	0.020839
Triggering receptor expressed on myeloid cells 1	*Trem1*	2.063924	0.023686
Tumor necrosis factor (ligand) superfamily, member 13	*Tnfsf13 (April)*	0.880437	0.046223
Ubiquitin specific peptidase 38	*Usp38*	−0.5067	0.035

**Table 2 life-12-00642-t002:** Genes and their relative mRNA expressions that are changed significantly after chronic tBHQ exposure and that are relevant in SARS-CoV-2 infection.

Gene Name	Gene Symbol	Fold Change	adjP
Cathepsin B	*Cts B*	0.514308	0.040677
Cd14 antigen	*Cd14*	1.154081	0.017954
Cd151 antigen	*Cd151*	0.449516	0.047853
Cd209d antigen	*Cd209d*	−1.78035	0.023051
Interferon induced transmembrane protein 1	*Ifitm1*	1.154081	0.021632
Interferon induced transmembrane protein 2	*Ifitm2*	1.174042	0.017954
Interleukin 8 receptor, beta	*Cxcr2 (Il8rb)*	2.650023	0.016262
Oncostatin M	*Osm*	0.748042	0.02052
Peptidyl arginine deiminase, type IV	*Padi4*	1.156019	0.040966
Perforin 1	*Prf1*	0.69257	0.03873
Protein phosphatase 1 regulatory subunit 3D	*Ppp1r3d*	2.300879	0.020477
Synapsin II	*Syn2*	1.057722	0.033471

## Data Availability

Not applicable.

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
