# Peer review of "Chronic Exposure to the Food Additive tBHQ Modulates Expression of Genes Related to SARS-CoV-2 and Influenza Viruses"

_life, 2022, doi:10.3390/life12050642_

Round 1

Reviewer 1 Report

“Chronic exposure to the food additive tBHQ modulates expression of genes related to SARS-CoV-2 and influenza viruses” is an interesting paper, well written, very clear. It is an important topic, as food additives are of concern, as they are present in so many products of human use.

Results are well presented, very objective. No doubts arise.

Reviewer 2 Report

The manuscript under review is an interesting study on the SARS-CoV-2 related genes expression in chronic exposure to tBHQ. The Authors found that 20-day-long exposure to tBHQ provoke a differential gene expression in mice. In particular, the altered expressed genes are those involved in the immune response to respiratory viruses (influenza and SARS-CoV-2). The study is well-conducted, and the methods are well explained. The discussion and conclusion are consistent with the results. I have only a few minor comments:

In the text, there are some English mistakes, for example:

  • Pasta is uncountable, so in the introduction, use the word “pasta” instead of “pastas”;
  • In the introduction, “tBHQ-dependent changes…suggest” instead of “suggests”;
  • In the discussion, “… tBHQ binds to HA proteins (hemagglutinin) of influenza viruses and stabilize the non-infectious conformation” stabilize should be stabilizes.

Please, make a whole manuscript English revision and correct the errors.

In the discussion, “tBHQ is a chemical that activates two major xenosensors (AHR and NRF2) and many genes, that are regulated by tBHQ have responsive elements that bind these receptors” this sentence is confused, please re-write it.

Figure 2 A is tiny, the words are difficult to read, is it possible to enlarge it a little bit?

Figure 3, all circles are concentrated in the lower y values, I suggest using a smaller negative log10 scale, for example 0-0.5-1-1.5-2… instead of 0-2-4 (this suggestion is not mandatory)

I suggest implementing the reference list with the following papers: 10.1016/j.brainres.2020.146818, 10.3390/medicina58020144, 10.1177/0025817220926915, 10.3390/antiox9080659, 10.3390/antiox10091491

Overall, I think this work is very good and deserves to be published.

Reviewer 3 Report

In this study, Nemeth et al. suggested that tBHQ, a common food additive could modulate virus-dependent processes and disease outcome in both influenza and SARS-CoV2 infections.

  1. The manuscript is interesting but needed a thorough English language revision for better understanding especially some sentences are divided into different part using comma (,) which can be write in direct form and make more good sense of understanding.
  2. Author used different genes but does not write anything about ACE2 which are major target of human infection causing CoVs including HCoV-NL63, SARS-CoV and SARS-CoV2.
  3. Author can also do molecular docking or report the binding capabilities of tBHQ as a ligand.
  4. Authors should have to rewrite the abstract and conclusion parts as in conclusion can write directly the finding and applications only.
  5. If it is possible, Authors can add a schematic diagram in discussion part like section 4.2 which can better illustrate the gene regulatory effect of tBHQ that can better depict that at which step or stage in immune system (receptors or antiviral role) the tBHQ have its functions
  6. Author should add high quality figure with visible figure labelling. Otherwise, should add the note at bottom of the figure what does abscissa or ordinate means.

Author Response

This manuscript is a resubmission of an earlier submission. The following is a list of the peer review reports and author responses from that submission.